# Targeting Protein Synthesis in Colorectal Cancer

**DOI:** 10.3390/cancers12051298

**Published:** 2020-05-21

**Authors:** Stefanie Schmidt, Sarah Denk, Armin Wiegering

**Affiliations:** 1Department of Biochemistry and Molecular Biology, Theodor Boveri Institute, Biocenter, University of Würzburg, 97074 Würzburg, Germany; stefanie.schmidt2@biozentrum.uni-wuerzburg.de (S.S.); sarah.denk@uni-wuerzburg.de (S.D.); 2Department of General, Visceral, Transplant, Vascular and Pediatric Surgery, University Hospital Würzburg, 97074 Würzburg, Germany; 3Department of Biochemistry and Molecular Biology, Comprehensive Cancer Center Mainfranken, University of Würzburg, 97074 Würzburg, Germany

**Keywords:** colorectal cancer, protein synthesis, translation initiation

## Abstract

Under physiological conditions, protein synthesis controls cell growth and survival and is strictly regulated. Deregulation of protein synthesis is a frequent event in cancer. The majority of mutations found in colorectal cancer (CRC), including alterations in the WNT pathway as well as activation of RAS/MAPK and PI3K/AKT and, subsequently, mTOR signaling, lead to deregulation of the translational machinery. Besides mutations in upstream signaling pathways, deregulation of global protein synthesis occurs through additional mechanisms including altered expression or activity of initiation and elongation factors (e.g., eIF4F, eIF2α/eIF2B, eEF2) as well as upregulation of components involved in ribosome biogenesis and factors that control the adaptation of translation in response to stress (e.g., GCN2). Therefore, influencing mechanisms that control mRNA translation may open a therapeutic window for CRC. Over the last decade, several potential therapeutic strategies targeting these alterations have been investigated and have shown promising results in cell lines, intestinal organoids, and mouse models. Despite these encouraging in vitro results, patients have not clinically benefited from those advances so far. In this review, we outline the mechanisms that lead to deregulated mRNA translation in CRC and highlight recent progress that has been made in developing therapeutic strategies that target these mechanisms for tumor therapy.

## 1. Introduction

Colorectal cancer (CRC) is the third-most common type of cancer and one of the leading causes of cancer-related deaths worldwide [1,2]. The standard of care for CRC includes surgical resection and neo/adjuvant chemotherapy. Although the five-year survival rate of patients with metastasized CRC has improved over the last few years due to resection of metastases and targeted treatment therapies, it is still below 20% [3,4].

The development of CRC is characterized by a defined spectrum of genetic changes, known as the adenoma-carcinoma sequence [5]. These specific alterations affect a plethora of cellular processes including cell proliferation, survival, stemness, metabolism, replication, invasion, and protein synthesis [6,7]. Among these, changes in the regulatory mechanisms of mRNA translation have gained more and more attention during the past few years. Dysregulation of these mechanisms is frequently observed in a variety of tumor entities, including CRC, and many links between oncogenic alterations and the translation machinery have been established [8,9,10]. Accordingly, the rates of protein synthesis are generally enhanced in malignant intestinal tissues as compared to normal tissues, making protein synthesis an attractive target for anti-CRC therapy [11,12,13].

In this review, we provide a mechanistic overview of deregulated mRNA translation in CRC and highlight promising targeting strategies as well as clinical advances for the treatment of this cancer.

## 2. Mechanisms of Regulation of Protein Synthesis

Protein synthesis, including ribosome biogenesis, is one of the most energy-intensive processes in living cells and is strictly regulated [14]. mRNA translation can be divided into four steps—initiation, elongation, termination, and ribosome recycling—each of which has a characteristic requirement for eukaryotic translation factors [15]. Here, we mainly focus on the process of initiation, as this is well-studied for deregulated activity in cancer (Figure 1).

### 2.1. Translation Initiation

Translation initiation starts with the formation of a ternary complex (TC) consisting of guanosine triphosphate (GTP)-bound eukaryotic initiation factor (eIF) 2 (eIF2-GTP) and initiator Met-tRNA (Met-tRNA_i_^Met^), which is delivered to the 40S ribosome. The TC-40S interaction is stabilized by eIFs 1, 1A, 3, and 5, forming the 43S pre-initiation complex (PIC), ready for mRNA binding [15,16,17,18]. Under physiological conditions, initiation of mRNA translation is then mediated by the 5′–7-methylguanosine cap, which is bound by the eIF4F complex comprised of eIFs 4A, 4E, and 4G, providing a binding site for the 43S PIC [19]. eIF4G functions as a scaffold, binding eIFs 4E and 4A, poly(A)-binding protein (PABP), and eIF3 [15,20]. PABP binds the 3′ poly(A) tail of mRNAs, and additional binding sites in PABP promote further protein–protein interactions, thereby influencing translation and mRNA metabolism [21,22].

Once the 43S PIC is loaded onto the mRNA, the ribosome and associated factors migrate along the 5′ untranslated region (UTR) until the Met-tRNA_i_^Met^ anticodon pairs with a suitable AUG start codon, leading to the formation of a stable 48S complex and hydrolysis of eIF2-bound GTP [23,24]. This step is mediated by the eIF2-specific GTPase-activating protein eIF5, which binds eIF2 and activates its GTPase activity. GTP hydrolysis triggers the partial dissociation of eIF2-GDP from 40S subunits [25]. For subsequent rounds of initiation, eIF2 is recycled by the eIF2B complex, which mediates guanine nucleotide exchange.

The last stage of initiation requires joining of the 60S ribosomal subunit to assemble an elongation-competent 80S ribosome [26]. This occurs simultaneously along with the dissociation of eIFs 1, 1A, and 3 and the remaining eIF2-GDP, which is mediated by the ribosome-dependent GTPase eIF5B [27].

### 2.2. Regulation and Alternative Pathways of Initiation

One of the most important steps during translation initiation is the formation of the eIF4F complex [15]. In particular, eIF4E and the phosphorylation status of its binding partners, the eIF4E binding proteins (4E-BPs), are limiting for the initiation process [28]. Hyperphosphorylation of 4E-BPs decreases their capacity to bind eIF4E, whereas hypophosphorylation enhances the binding capacity and prevents formation of an active eIF4F complex [29]. Furthermore, recycling of the TC, which is mediated by the eIF2B complex, is a rate-limiting step of translation initiation. However, under stressful conditions, eIF2α can be phosphorylated at S51 by the stress-related kinases GCN2, HRI, PERK, or PKR; this is commonly known as the integrated stress response (ISR) [30,31]. Phosphorylation at this site enhances the affinity of eIF2 to bind eIF2B, leading to the sequestration of this complex and reduction of the nucleotide exchange function of eIF2B. Consequently, cap-dependent translation is impaired and alternative pathways of translation initiation are promoted, including upstream open reading frame (uORF)–mediated translation of stress-related genes or initiation at internal ribosome entry sites (IRES) [31].

### 2.3. Elongation and Termination

Upon 80S complex formation, the ribosome is primed for translation elongation [32]. Once the Met-tRNA_i_^Met^ anticodon is located at the peptidyl (P) site of the ribosome, it base-pairs with the start codon of an mRNA [33]. The eukaryotic elongation factor (eEF) 1A binds aminoacyl-tRNA and delivers it to the acceptor (A) site of the ribosome, in which the second codon of the open reading frame is present. Upon codon recognition, aminoacyl-tRNA is incorporated into the A site. Subsequently, a peptide bond is formed with the tRNA located in the P site and ratcheting of the ribosomal subunits triggers eEF2-promoted movement of tRNAs from P and A sites to exit (E) and P sites, respectively [32]. Translation is terminated when the ribosome reaches the end of the coding region and a stop codon enters the A site, followed by peptide release and ribosomal subunit dissociation.

## 3. The Adenoma-Carcinoma Sequence and Its Impact on Deregulation of Protein Synthesis in CRC

The majority of CRCs arise from benign adenomas that develop into a malignant carcinoma, a process that is characterized by accumulation of specific mutations, the so-called adenoma-carcinoma sequence (Figure 2) [34]. In over 95% of the cases, the initiating event is a mutation in the WNT signaling pathway, of which 80% are mutations in the *APC* tumor suppressor gene [5,35]. Non-functional APC leads to hyperactive WNT signaling, resulting in deregulated expression of WNT target genes [6]. Among these, the *MYC* oncogene is an essential driver of colorectal tumorigenesis, and *MYC* deletion rescues intestinal hyperproliferation induced by loss of APC in vivo [36]. MYC drives the transcription by all three RNA polymerases, thereby controling essential cellular processes including ribosome biogenesis and protein synthesis [37,38,39,40,41]. Accordingly, overexpression of ribosomal proteins (RPs) and enhanced ribosome biogenesis in general has been established as an early event during CRC tumorigenesis [42]. APC deficiency, via MYC upregulation, regulates many more genes associated with translation and is also implicated in balancing the cellular responses to stress signaling by influencing activities of stress-related kinases and the eIF2α/eIF2B complex [12]. It is likely that this latter mechanism is necessary for fine-tuning the rates of protein synthesis and the stress response in CRC cells throughout all adenoma-carcinoma stages to ensure tumor cell survival. Further investigations are needed to shed light on potential therapeutic implications. Another rate-limiting translation initiation factor regulated by the APC-MYC axis is eIF4E. Correspondingly, its overexpression takes place in the early adenoma stage, but nevertheless also correlates with late tumor stages and metastasis [43,44,45].

In addition to MYC, two other oncogenic pathways—RAS/MAPK and PI3K/AKT—are master regulators of protein synthesis and are frequently deregulated in CRC [35,46]. Alterations in these pathways occur later between the early and late adenoma stages. Therefore, it is not surprising that increases in the levels of p-mTOR, p-p70-S6K1, and p-4E-BPs were found to be associated with metastasis, which is the late event finally leading to an invasive carcinoma [47,48,49]. Besides upregulation of mTORC1 activity via these later events, APC deficiency has also been shown to directly increase mTORC1 signaling [50]. All in all, these essential associations between the genetic alterations in the course of the adenoma-carcinoma sequence and deregulation of the translation machinery underscores a fundamental role for enhanced protein synthesis rates in controling both the initiation and progression of CRC.

## 4. Deregulation of Protein Synthesis in CRC and Potential Therapeutic Strategies

In this section, we summarize current knowledge about regulatory factors and mechanisms involved in mRNA translation and how they are deregulated in CRC (Figure 1, and Table 1). Furthermore, examples of targeting possibilities and their applicability in CRC are outlined.

### 4.1. Deregulation of Ribosome Biogenesis in CRC

Ribosomes are supramolecular RNA–protein complexes with a high degree of sequence and structure conservation among bacteria, eukaryotes, and archaea [32]. The human ribosome comprises 80 ribosomal proteins as well as four ribosomal RNAs (rRNAs)—5S, 5.8S, 18S, and 28S rRNA [51]. First reports pointing to deregulated ribosome biogenesis in CRC reach back to the 1980s, where it was shown that genes encoding RPs are overexpressed in human CRC cell lines and in samples of human colon carcinoma as compared to normal mucosa [52,53,54]. Since then, several studies have validated the deregulated expression of certain RPs in CRC, and also knockdown experiments, e.g., of RPS24 or RPL15, showed that a decrease in RP levels reduces proliferation and migration and induces apoptosis in CRC cells [42,55,56,57]. Therefore, the increased abundance of RPs can enhance ribosome biogenesis and protein synthesis subsequently driving oncogenic transformation in CRC. This is consistent with reports showing an upregulation of mRNAs and proteins associated with ribosome biogenesis in stem cell-enriched murine intestinal organoids [58]. In contrast, mutation and loss of specific RP function is linked to microsatellite-unstable (MSI) CRC and diseases with high susceptibility to CRC development, such as ribosomopathy Diamon–Blackfan anemia (DBA) and familial CRC type X (FCCX), a hereditary nonpolyposis microsatellite-stable (MSS) CRC [59,60,61].

Genes encoding RPs as well as factors necessary for rRNA processing and transport of ribosomal subunits are transcribed by RNA polymerase II; this process is highly coordinated by the MYC oncoprotein. Besides RNA polymerase II, RNA polymerases I and III are also involved in translational regulation via the transcription of rRNA. As MYC induces transcription by all three RNA polymerases, it stimulates rRNA synthesis [38,39,40,62,63]. MYC is almost universally amplified or overexpressed in CRC due to APC mutation, which is an early event in CRC tumorigenesis (Figure 2, see also Section 3) [6]. Consistent with this, a computational study of the MYC expression network from the Cancer Genome Atlas (TCGA) showed that high MYC expression in colon and rectum adenocarcinoma positively correlates with pathways associated with translation, ribosomes, and rRNA [64]. Regarding ribosome biogenesis, enhanced rRNA synthesis in CRC cells is driven by MYC, as MYC depletion reduces nascent levels of 5S, 5.8S, 18S, and 28S rRNAs, which is connected with decreased translation rates [65]. More specifically, there is compelling evidence that rRNA synthesis is hyperactivated in CRC due to APC deficiency in general. CRC cells lacking functional APC activate upstream binding factor (UBF), a factor necessary for rDNA transcription, thereby enhancing the expression of pre-45S rRNA in human CRC tissue samples and cell lines, which correlates with poor survival [66]. Furthermore, recent studies validated that APC deficiency in CRC upregulates the expression of genes encoding RPs and auxiliary factors of ribosome assembly, and enhances protein synthesis rates [11,12,66].

The second-most common mutation in CRC is found in KRAS, leading to constitutively active MAPK signaling, which is also implicated in the regulation of rDNA synthesis [67]. Additionally, CRC cells with mutant KRAS show increased expression of genes involved in ribosome biogenesis and mRNA translation; this is associated with enhanced protein synthesis in CRC cells as compared to cells with wild-type KRAS [68]. Finally, the mTOR signaling pathway is a master regulator of ribosome biogenesis and is highly deregulated in CRC [69]. Mechanisms of deregulation of the mTOR pathway are discussed in Section 4.2.

#### Targeting Ribosome Biogenesis in CRC

CRC is commonly treated using a combination of several chemotherapeutics (FOLFOX, FOLFIRI) including 5-fluorouracil (5-FU) and oxaliplatin [4]. Interestingly, besides eliciting a DNA damage response, both drugs affect ribosome biogenesis, particularly in CRC [70,71,72,73,74]. In addition to blocking DNA synthesis, 5-FU can be incorporated into all RNA species, thereby altering RNA metabolism. Accordingly, 5-FU leads to reduction of protein synthesis, possibly due to reduced ribosome biogenesis, as well as translational reprogramming of specific mRNAs in CRC cells [71]. Oxaliplatin belongs to the group of platinum-containing chemotherapeutics and is a derivative of cisplatin. Oxaliplatin treatment of CRC is widely established and it has a different side-effect profile as compared to cisplatin, suggesting that oxaliplatin has a different mechanism of action [75,76]. Consistent with this, the efficacy of oxaliplatin is not dependent on its ability to induce a DNA damage response, but rather to induce ribosome biogenesis stress [72]. This, in turn, deprives the cells of translation machinery components, on which CRC cells are highly dependent. These data imply that certain approved chemotherapeutic drugs could be used in a more mechanism-based manner for the treatment of CRC, however, further investigation is needed in this direction.

A more targeted approach for interfering with ribosome biogenesis is the development of specific RNA polymerase I inhibitors, namely CX-3543 and CX-5461 [77,78,79]. Both these inhibitors bind to G-quadruplex structures enriched in rDNA genes and inhibit the binding of certain co-factors and transcription of rDNA. With respect to their anti-cancer effects, they are efficacious in vitro as well as in vivo and are in phase I/II clinical trials for several hematological and solid malignancies [80,81]. Several studies have investigated the anti-cancer activities of CX-5461 and CX-3543 in both CRC cells and xenograft models [77,82,83,84]. Interestingly, sensitivity to both compounds is dependent on the DNA damage response proteins BRCA1/2 [83]. This implies that these inhibitors can be used for treating cancer entities having defects in DNA damage repair, which is a common feature of MSI CRC [84]. Moreover, CX-5461 not only affects CRC cell viability and xenograft growth, but also stem cell properties and differentiation of murine intestinal organoids [85]. However, it requires further clarification on how exactly ribosome biogenesis contributes to stem cell features and intestinal tumorigenic transformation.

### 4.2. Deregulation of mTOR Signaling and Translation Factors in CRC

One major regulator of translation is the mTOR1 complex 1 (mTORC1), whose activity is primarily controlled by the RAS/MAPK and PI3K/AKT pathways [46]. KRAS and PI3K/AKT are frequently mutated or overexpressed in CRC, leading to hyperactivation of mTORC1, either indirectly via MEK-ERK-RSK-Raptor phosphorylation in the case of KRAS or directly in the case of PI3K/AKT (Figure 1) [6,69,86,87]. In addition, about 10% of patients with metastatic CRC carry BRAF mutations, in particular the V600E mutation, which is associated with poor prognosis [88]. Mutation of BRAF leads to increased phosphorylation and activity of this protein kinase and, in turn, sustained MAPK pathway activity. Furthermore, PTEN, a negative regulator of PI3K/AKT, is also deleted in CRC [6]. Besides deregulated upstream signaling pathways, mTOR itself is found to be mutated in CRC, with some of the mutations resulting in hyperactivation of mTORC1 [89,90]. Also, mRNA and protein levels of mTOR are increased in heterozygous mutant Apc^∆716^ mice, a model for human familial adenomatous polyposis (FAP), which is dependent on β-Catenin, suggesting a direct role of the WNT signaling pathway in mTOR regulation [50].

The main effectors of mTOR signaling are p70-S6K1 and 4E-BPs. mTORC1 directly phosphorylates 4E-BPs, thereby relieving the inhibitory binding of 4E-BPs to eIF4E (see also Section 2.2.) [29]. Levels of phosphorylated 4E-BPs are increased in CRC tissue as compared to normal mucosa, and this correlates with metastasis and poor prognosis, and 4E-BPs have been established as critical mediators of the tumourigenic properties of CRC [48,49,91]. The second effector of mTORC1 phosphorylation, p70-S6K1, targets the ribosomal protein S6 (rpS6), PDCD4, and eEF2K [92]. Phosphorylation of p70-S6K1 has been identified as a prognostic marker for CRC and is associated with poor survival [93]. p70-SK61-mediated phosphorylation of PDCD4 and eEF2K induces degradation or inactivation of these proteins [94,95]. Being an established tumor suppressor, PDCD4 is downregulated in CRC, suggesting enhanced eIF4A activity and translation initiation, since it usually binds and inhibits eIF4A [96,97,98,99]. When eEF2K is inactivated by p70-S6K1, the inhibitory phosphorylation of eEF2 is relieved, and translation elongation proceeds [100]. Correspondingly, eEF2K has tumor-suppressive functions in CRC, as depletion of eEF2K enhances cell survival in CRC and low expression levels of eEF2K in CRC patient tissue samples correlate with poor clinical outcome [101,102]. The role of eEF2K in the control of intestinal tumorigenesis has also been validated in vivo, showing that mTORC1-p70-S6K1-mediated inactivation of eEF2K enhances the activity of eEF2 in VillinCre^ER^Apc^fl/fl^ mice, which accelerates translation elongation and protein synthesis in intestinal tissue [11]. Due to its role in driving translation elongation, the substrate of eEF2K, eEF2, is overexpressed in gastrointestinal malignancies, including CRC, and its knockdown impairs the proliferative potential of CRC cells [103,104]. Although the role of other translation elongation factors in the development of CRC has not been well studied, a recent systematic analysis of Oncomine and TCGA datasets revealed deregulated expression of several elongation factors in CRC tissue, whereas the prognostic value of expression levels was variable [105].

It has long been known that eIFs are deregulated in cancer (see also for detailed review [8,106]). The first translation initiation factor to be identified as overexpressed and correlated with CRC progression and occurrence of metastasis was eIF4E [43,44,45,107]. In addition to upregulation of eIF4E, eIF4E phosphorylation at S209 is significantly higher in CRC tissue than corresponding non-tumorigenic tissue [108]. MNK1 and MNK2 are the two kinases that phosphorylate eIF4E, the only well-characterized substrate of these kinases [109,110]. The role of eIF4E phosphorylation is not completely understood, but it seems to be essential for the tumor-promoting functions of eIF4E [111]. Besides eIF4E, the RNA helicase eIF4A1 is involved in positive regulation of translation initiation. It facilitates synthesis of many proto-oncogenic mRNAs, such as the MYC mRNA, with long structured 5′ UTRs, which underlines the role of eIF4A1 in cancer development [112]. eIF4A1 expression is increased in CRC tissue and is directly regulated by miR-133a, which is downregulated in CRC [113]. Another member of the eIF4A family, eIF4A2, plays a role in miRNA-mediated translational repression; however, this function of eIF4A2 is debated [114,115,116]. eIF4A2 also promotes invasion of CRC cells as well as lung metastasis in xenografts and is associated with poor prognosis [117]. In addition, there are numerous reports on the deregulated expression of the majority of initiation factors in CRC, which will not be discussed in detail here [118,119,120,121,122,123,124,125,126].

### 4.3. Targeting mTOR Signaling and Translation Factors in CRC

As mTOR signaling is activated in the majority of cancers, several inhibitor compounds, inhibiting either PI3K or mTOR or both, have been developed during the last decade. Early studies suggested the interference of mTOR activity for treatment of CRC [127]. The first established specific mTORC1 inhibitor, rapamycin (sirolimus), and its analogues (rapalogues such as temsirolimus and everolimus) have been extensively studied with respect to their anti-tumorigenic effects in CRC in vitro and in vivo [50,69,86,128,129,130,131]. Unfortunately, single-agent partial mTOR inhibition has limited therapeutic benefits, leading to feedback activation of the PI3K/AKT pathway and drug resistance [132]. This led to the development of the dual PI3K/mTOR inhibitor, which targets the ATP-binding sites of both the kinases [133]. The impact of dual PI3K/mTOR inhibition on CRC has been reviewed in detail in [69,86,134,135]. One widely used dual inhibitor, NVP-BEZ235 reduced the viability of both mutant and wild-type PI3K catalytic subunit alpha (PI3KCA) cell lines, APC-deleted, PI3KCA-mutated organoids and delayed tumor growth in respective colon tumor models [136,137]. An approach by our group used BEZ235 to target MYC expression in CRC cells, as MYC turnover and translation are highly regulated by the PI3K and mTOR pathways, respectively [97]. Surprisingly, instead of reducing MYC protein levels, BEZ235 enhanced MYC expression due to elevated transcriptional and cap-independent translational upregulation. This was counteracted by the natural compound silvestrol, an eIF4A helicase inhibitor [138], which reduced MYC translation and intestinal tumor growth. In addition to silvestrol, a second natural eIF4A inhibitor, elatol, elicited anti-tumorigenic effects in CRC cell lines as well as in xenograft and PDX mouse models [117,139,140]. Besides natural compounds, the synthetic eIF4A inhibitor FL3 reduces the viability of CRC cell lines as well as tumor growth in CRC xenografts [141,142]. Regarding other subunits of the eIF4F complex, the small molecule 4EGI disrupts the interaction between eIF4E and eIF4G and stabilizes the association of eIF4E with 4E-BPs [143,144]. So far, there is limited data showing that 4EGI reduces the tumorigenic potential of mouse or human CRC cells [145,146]. The function of eIF4E can also be indirectly targeted by inhibiting MNK1/2-mediated phosphorylation at S209. This is an attractive approach, as MNK function and eIF4E phosphorylation are not required for normal development [147]. Besides the anti-fungal agent cercosporamide, which was identified as an MNK1/2 inhibitor reducing CRC cell viability as well as growth of CRC xenografts [148], the more specific MNK1/2 inhibitor eFT508 was identified through structure-based design [149]. Although eFT508 was weakly effective against the growth of CRC cells in vitro, it reduced tumor growth in a colon allograft model with comparable effects to a newly developed MNK1/2 inhibitor [150]. Additionally, as MYC drives mRNA translation of programmed death-ligand 1 (PD-L1) in KRAS-/MYC-driven liver cancer, which favors immune escape of the tumor, treatment of those mice with eFT508 or anti-PD-L1 therapy significantly prolonged the survival of the mice and reduced metastasis [151]. These data suggest that eFT508 may open a wide spectrum of clinical applications including CRC.

### 4.4. Deregulation of the Translation Initiation Factors eIF2/eIF2B and the ISR in CRC

Two other essential regulators of translation initiation are the eIF2 complex (comprised of eIF2α/β/γ), a part of the TC, and the eIF2B complex (comprised of two heterodimers of eIF2Bα/β/ γ/δ/ε) [152,153]. In addition to their role in controling initiation, the two complexes are involved in the cellular stress response, in particular the ISR, mediated by phosphorylation of eIF2α by four different kinases (see also Section 2.2) [154]. It is a common phenomenon that tumor cells encounter various stress stimuli, including oncogene activation, replicative stress, hypoxia, nutrient deprivation, and, therefore, have elevated p-eIF2α S51 levels [155,156,157]. The most studied eIF2α kinase is PERK, which is classically activated by the accumulation of misfolded proteins in the endoplasmic reticulum (ER), termed as “ER stress” [158]. On the one hand, active PERK/eIF2α signaling has been suggested to have oncogenic potential in CRC, whereas on the other hand, a more tumor-suppressive role of PERK in CRC has been documented [159,160,161,162,163]. Additionally, the PERK/eIF2α signaling node is involved in the regulation of intestinal stemness and differentiation, where ER stress is associated with loss of stemness in a PERK/eIF2α-dependent manner, defining a potential role of PERK/eIF2α in CRC development [164,165]. The second eIF2α kinase, GCN2, is mainly activated by amino acid deprivation, as it binds uncharged tRNAs [166]. Although there has been limited data on the influence of GCN2 on CRC tumorigenesis, nutrient deprivation in solid tumors, including CRC, enhances the activity of the GCN2/eIF2α pathway [12,167]. The third kinase, PKR, is activated by dsRNA binding in response to viral infection, although other mechanisms have also been described [168,169]. However, the contribution of PKR to tumorigenesis in general and to CRC development in particular is not well defined, with studies claiming both tumor-suppressive and oncogenic roles of PKR/eIF2α signaling in CRC [12,170,171,172,173,174,175].

Regarding the role of stress signaling in the development of CRC, it is interesting to note, that intestinal inflammatory diseases (intestinal bowel disease, IBD), such as Crohn’s disease (CD) or ulcerative colitis (UC), are also characterized by increased ER stress levels and ISR signaling [176]. Patients with IBD have a higher susceptibility to develop CRC, emphasizing the important influence of deregulated stress pathways in CRC [177].

With respect to eIF2α, the key component of the ISR, both unphosphorylated and phosphorylated forms are upregulated in CRC tissues as compared to normal mucosa, and elevated p-eIF2α S51 levels are found in the intestine of APC-deficient mice [12,178,179]. Enhanced eIF2α phosphorylation, mediated by GCN2, is an important oncogenic mechanism in APC-deficient cells that are characterized by elevated MYC and cellular stress levels in comparison to APC-proficient cells [12]. Phosphorylated eIF2α binds tightly to eIF2B and sequesters it in an inactive complex, thereby limiting the otherwise high translation rates and, in parallel, preventing dephosphorylation of eIF2α. Silencing of the eIF2B subunit eIF2B5 (eIF2Bε) uncouples this mechanism, which finally leads to elevated protein synthesis and stress signaling, thereby driving MYC-induced apoptosis. This establishes an essential dependency of APC-deficient cells on eIF2B5. However, there is limited data describing the potential oncogenic role of eIF2B in general, and detailed studies in CRC are lacking [180,181].

### 4.5. Targeting the Translation Initiation Factors eIF2/eIF2B and the ISR in CRC

The aforementioned data point to potential therapeutic strategies for CRC by targeting the p-eIF2α/eIF2B signaling node. Both eIF2 and eIF2B play an essential role in regulating mRNA translation. There is a clear therapeutic window for targeting eIF2B; however, further studies are needed to determine how exactly the activity or formation of the p-eIF2α/eIF2B complex has to be modulated to specifically act on CRC cells [12]. Moreover, to date, there are no direct inhibitors available, such as inhibitors of eIF2B5 GEF activity, disrupting the catalytic function of eIF2B, or small molecules that inhibit the p-eIF2α/eIF2B interaction. One compound that interferes with p-eIF2α-mediated ISR is the integrated stress response inhibitor (ISRIB). ISRIB was originally identified as an inhibitor of the downstream effects of eIF2α phosphorylation during an ISR, and to stabilize the eIF2B decamer, thereby enhancing GEF activity [152,182,183,184,185,186,187,188]. ISRIB has anti-tumorigenic effects in aggressive prostate cancer and can be used in vivo; however, studies investigating the action of ISRIB on CRC are lacking [189,190,191,192]. Another strategy to target eIF2α phosphorylation is the small molecule-mediated inhibition of its phosphatase complex, with protein phosphatase 1 (PP1) as the catalytic core protein, which dephosphorylates eIF2α to terminate the ISR [193,194,195,196]. Interestingly, salubrinal, a specific inhibitor of PP1 [197], elicits reversible differentiation of intestinal stem cells via activation of an unfolded protein response, resulting in increased sensitivity to oxaliplatin treatment in CRC xenografts [198].

As discussed above, activation of eIF2α kinases and ISR have oncogenic properties in certain tumor settings. Therefore, inhibition of these kinases has been thought to be a promising anti-cancer approach. In contrast, inhibition of PERK by the small molecule inhibitor GSK2656157, or the related GSK2606414, does not specifically affect the viability of APC-deficient CRC cells or murine intestinal organoids [12,199,200]. Inhibitor compounds for both GCN2 and PKR have also been developed [201,202,203,204,205]. The first commercially available GCN2 inhibitor A-92, a triazolo [4,5-d] pyrimidine derivative, reduces the viability of APC-deleted CRC cells as well as murine and patient-derived organoids (PDOs), however, it cannot be used in vivo [12]. In parallel experiments, the potent imidazolo-oxindole PKR inhibitor C16 induced cell death to a similar degree and with similar specificity, but the mechanism was not well-defined. Despite its high potency and suitability for in vivo use, C16 lacks specificity, as it also targets cyclin-dependent kinases [206]. Recently, approaches for developing optimized GCN2 inhibitors, including compounds that contain the triazolo [4,5-d] pyrimidine core, have been investigated; some of these small molecules exert anti-proliferative activity in CRC cells [202,203,204]. In contrast, treatment with other GCN2 inhibitors alone had no effect on cell viability but sensitized a panel of CRC cell lines to asparaginase treatment, which is usually used as an anti-leukemic substance [203,207,208]. Besides the potential lack of specificity and unsuitability for in vivo use, a major drawback of interfering with eIF2 kinase function is their redundancy, with GCN2 and PERK being able to compensate for each other [12,209,210,211,212,213,214].

In summary, more detailed studies are needed to provide clarifications regarding the conditions under which the eIF2/ISR signaling pathways have pro-survival or pro-apoptotic roles in CRC and to carefully evaluate possible therapeutic strategies.

## 5. Clinical Advances in Targeting Protein Synthesis in CRC

Over the last decade, several tyrosine kinase inhibitors as well as checkpoint inhibitors for immunotherapies have been approved for the treatment of patients with advanced or relapsed CRC. Among these, only a few target protein synthesis mechanisms. In this section, we highlight existing clinical data and summarize ongoing clinical trials.

### 5.1. Clinical Data for Targeting mTOR Signaling in CRC

As discussed in Section 4.2 and Section 4.3, the mTOR pathway is a reasonable therapeutic target in CRC. Although most of the inhibitors show promising results in vitro and in vivo, clinical trials have mainly failed [136,215].

#### 5.1.1. mTORC1 Inhibitors—Rapalogues

In a phase II clinical trial, the therapeutic potential of everolimus was investigated in patients with metastatic CRC previously treated with bevacizumab, and fluoropyrimidine-, oxaliplatin-, and irinotecan-based regimens (NCT00419159) [216,217]. Overall, 199 patients were enrolled, who received either 70 mg/week or 10 mg/day of everolimus, of which 71 patients per group were included in the per protocol analysis. The best overall achieved response was stable disease in 45 out of 142 patients (31% in the 70 mg/week group and 32.4% in the 10 mg/day group). The median duration of stable disease was 3.9 months in both groups, whereas overall survival was 4.9 months in the weekly group and 5.9 months in the daily group. Specifically, patients with a KRAS mutation had a reduced overall survival.

In a second phase II trial, patients with KRAS-mutated and chemotherapy-refractory CRC were treated with temsirolimus until tumor progression, and with a combination of temsirolimus and irinotecan from this point onwards (NCT00827684) [218]. The median time until tumor progression was 45 days with temsirolimus monotherapy and 84 days with combination therapy. Patients receiving monotherapy showed no response despite 38% of the patients showing stable disease, whereas 63% of the patients receiving combination therapy showed stable disease. Furthermore, low plasma levels of KRAS were associated with a significantly enhanced prognosis.

The combination of rapamycin therapy and a preoperative 5 × 5 Gy radiotherapy in rectal cancer patients was also analyzed [219]. In part I dose escalation study, an unexpected high rate of postoperative toxicity for surgery was observed three days after the last dose of rapamycin. In part II of the trial, the time frame between the last dose of rapamycin and surgery was prolonged to 50 days. The percentage of pathological complete response and good responses (pT1), as expected from other trials, were 3% and 10%, respectively. The metabolic activity of tumors was significantly downregulated in PET-CT imaging after rapamycin treatment.

#### 5.1.2. Single PI3K and Dual PI3K/mTOR Inhibitors

Buparlisib (BKM120) is a pan-class I PI3K inhibitor with an IC_50_ in the low nanomolar range [220]. Two phase I trials were carried out to evaluate the tolerable dose. One trial included only patients with mutations in the PI3K pathway, and the other trial was performed in combination with irinotecan (NCT01833169, NCT01304602) [221,222]. In both trials, BKM120 was well-tolerated by patients but did not show any obvious therapeutic benefit. Moreover, an ongoing trial (NCT01591421) will evaluate the impact of a combined treatment of BKM120 and panitumumab, a monoclonal antibody targeting the epidermal growth factor receptor (EGFR), in recurrent or metastatic CRC with wild-type KRAS [223].

Alpelisib is a selective oral inhibitor of the PI3K catalytic subunit p110α [224]. In 2019, after positive results were obtained in the placebo-controlled SOLAR-1 trial (NCT02437318), it was approved by the FDA for the treatment of hormone receptor (HR)-positive, human epidermal growth factor receptor 2 (HER2)-negative, PIK3CA-mutated, advanced, or metastatic breast cancer [225]. Results of the dose escalation study suggested the therapeutic potential of alpelisib in CRC patients, with a disease control rate of 34.3% [226]. A phase Ib dose escalation study demonstrated that alpelisib in combination with a RAF kinase inhibitor and a monoclonal antibody targeting EGFR in BRAF-mutated CRC patients showed promising results (NCT01719380) [227]. For the ongoing subsequent phase II trial, an interim analysis of 102 patients suggested that alpelisib may have a progression-free survival benefit [228].

Finally, the dual PI3K/mTOR inhibitor GDC-0980 was evaluated in combination with 5-FU–based chemotherapy in advanced CRC in a phase Ib trial, and a partial response in two out of 19 patients was observed [229,230].

### 5.2. MNK1/2 Inhibitors in CRC

Currently, several phase I and II clinical trials evaluating the therapeutic potential of the MNK1/2 inhibitor eFT508 with or without anti-PD-L1 therapy in various tumor types have been initiated or already completed (clinicaltrials.gov). Of those, an already completed phase I/II trial (NCT03258398) evaluated a combination therapy of eFT508 and avelumab, a human monoclonal antibody targeting PD-L1, in patients with MSS CRC [231]. Experimental part 1 focused on evaluating the efficient dose of eFT508 with a fixed dose of avelumab. In experimental part 2, the defined dose of eFT508 from part 1 was tested with or without avelumab. The readout is the overall response rate in a time frame of 8–16 weeks. The results of the experimental part 1 were published on the American Society of Clinical Oncology (ASCO) conference in 2019. Of note, one patient from part 1 of the study showed a partial response for more than 8 months. Further data on clinical responses are expected in 2020.

Another selective MNK1/2 inhibitor, BAY1143269, has been described to have strong anti-tumor activity in vitro and in vivo [232]. Besides one phase I dose-finding trial (NCT02439346), no data on the activity of this compound have been published so far.

### 5.3. Antisense Oligonucleotides Against eIF4E

As described in Section 4.2., eIF4E is deregulated in a vast majority of tumor entities including CRC. Duffy et al. (2016) evaluated the impact of the antisense oligonucleotide ISIS 183750 targeting eIF4E in a clinical trial including mainly patients with CRC (NCT01675128) [233]. Of the 15 patients with irinotecan-refractory cancer, none showed a partial response, though seven (47%) had a stable disease. The median progression-free survival was 1.9 months, and the median survival was 8.3 months. Moreover, the peripheral blood samples of 13 out of 19 patients showed reduced eIF4E mRNA levels. Though antisense oligonucleotides were detected in the tissue of all patients, eIF4E protein levels did not change in CRC tissue.

## 6. Conclusions

The majority of the regulatory mechanisms of protein synthesis is deregulated in CRC. Over the last decade, many factors and signaling pathways that contribute to balanced mRNA translation have been investigated for their suitability as therapeutic targets in CRC (Figure 1). There is a tremendous amount of data showing promising results in pre-clinical studies of strategies targeting ribosome biogenesis, pathways regulating translation initiation, translation initiation factors, and translational responses to stress. However, only a few of these strategies advanced into clinical trials, and of those, most revealed only limited efficacy. Similar to what has been observed in other cancer types, in future, it is necessary to carefully analyze whether inhibition of protein synthesis is the key targeting strategy for CRC or there might be more efficient strategies to exploit the adaptive responses of cancer cells in protein synthesis. Consistent with this, as protein synthesis is an essential and general cellular process, there is a need to develop therapeutics that 1) have discrete targets within the translational machinery, and 2) affect molecules or pathways that are specifically altered in intestinal tumor cells, but not in non-malignant cells and tissues. One promising approach in this direction, that has gained attention during recent years, is the establishment of intestinal PDO biobanks [234,235,236,237]. These biobanks contain intestinal organoids of a large number of patients, that faithfully resemble the genomic landscapes of the original tumors. Therefore, they can facilitate the understanding of the molecular events underlying CRC progression, including deregulation of the translational machinery. Furthermore, they are valid tools for drug screens to not only identify potential new targets of established drugs associated with mRNA translation but also drug combinations of translation modulators that could be more efficient than single targeting. In sum, this would clearly open the opportunity to extend therapeutic options for CRC patients.

## Figures and Tables

**Figure 1 cancers-12-01298-f001:**
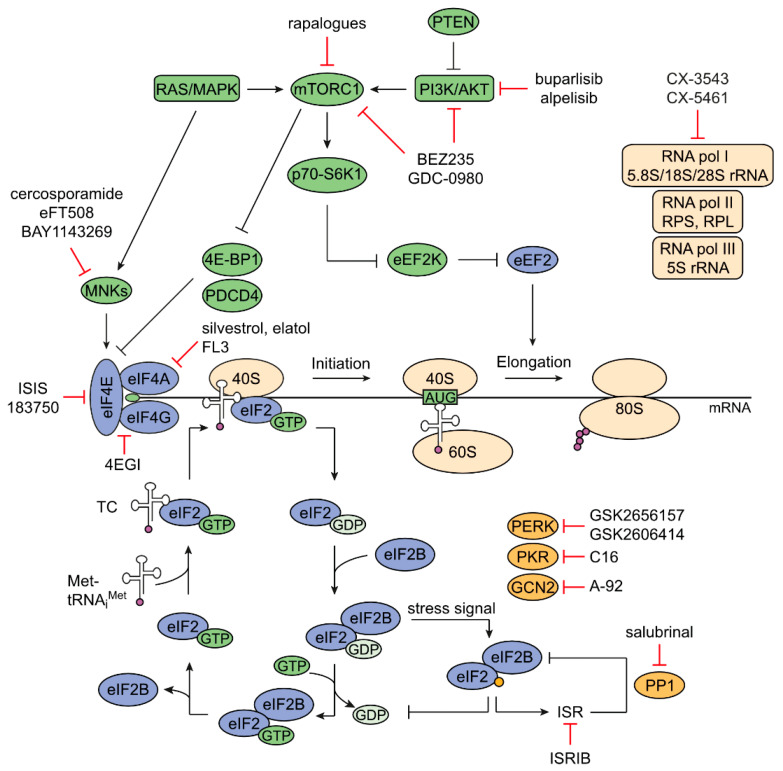
Schematic overview of regulation of mRNA translation and targeting possibilities in CRC. A large number of translation factors, signaling pathways, and ribosomal components are involved in the regulation of mRNA translation in general, and in particular in CRC. Different targeting strategies for interfering with deregulated protein synthesis have been developed as potential therapeutics, but clinical efficacy has been limited so far. For simplicity, only translation factors and signaling proteins are shown, which are described as potential therapeutic targets in the text (discussed in detail in Section 4). Black lines with arrow: activating signal; black lines with T bar: inhibitory signal; red lines with T bar: inhibition by small molecules or other substances; violet dots: amino acids generating a polypeptide chain, yellow dot: phosphorylation; green dot: 7-methylguanosine cap of mRNA; TC: ternary complex; ISR: integrated stress response; RNA pol I–III: RNA polymerase I–III.

**Figure 2 cancers-12-01298-f002:**
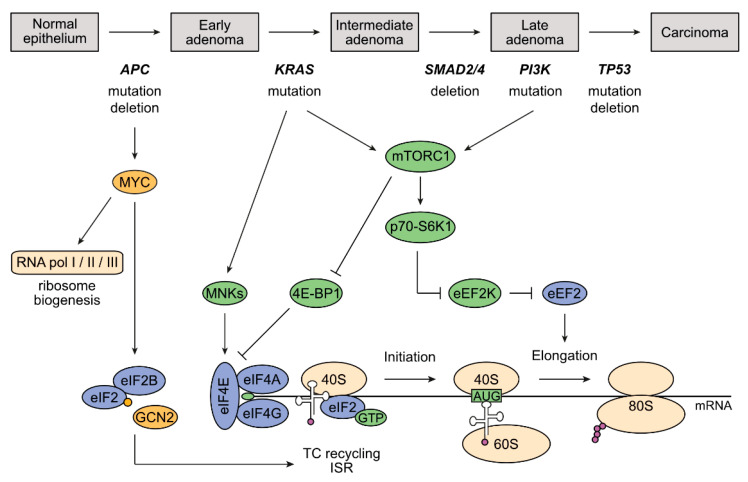
Genetic alterations in CRC in the adenoma-carcinoma sequence and their influence on mRNA translation. CRC develops over a series of clearly defined stages that are characterized by specific changes in oncogenes and tumor suppressor genes, that in turn regulate diverse mechanisms involved in mRNA translation. Black lines with arrow: activating signal; black lines with T bar: inhibitory signal; violet dots: amino acids generating a polypeptide chain, yellow dot: phosphorylation; green dot: 7-methylguanosine cap of mRNA; TC: ternary complex; ISR: integrated stress response; RNA pol I-III: RNA polymerase I–III.

**Table 1 cancers-12-01298-t001:** Deregulated factors and pathways in CRC.

Regulators of mRNA Translation		Deregulation in CRC	Impact on mRNA Translation
Ribosomal Components	RPL15	upregulation	enhanced ribosome biogenesis
	RPL22	mutation, downregulation	potentially deregulated translation of pro-apoptotic proteins and metastasis-related proteins
	RPS20	mutation	defect in pre-ribosomal RNA maturation
	RPS24	upregulation	enhanced ribosome biogenesis
	ribosomal RNAs	upregulation via MYC-mediated deregulation of RNA pol I and III activity	enhanced ribosome biogenesis
Signaling Pathways and Associated Factors	RAS/MAPK signaling	mutation and hyperactivation	hyperactivation of mTORC1 and subsequent activation of p70-S6K1 and inhibition of 4E-BPs leading to enhanced translation initiation
	PI3K/AKT signaling	mutation and hyperactivation, upregulation	hyperactivation of mTORC1 and subsequent activation of p70-S6K1 and inhibition of 4E-BPs leading to enhanced translation initiation
	PTEN	deletion	upregulation of PI3K/AKT signaling
	mTORC1	mutation and hyperactivation, overexpression, increased phosphorylation of mTOR	activation of p70-S6K1 and inhibition of 4E-BPs leading to enhanced translation initiation
	4E-BPs	increased phosphorylation	release of eIF4E and enhanced translation initiation
	PDCD4	downregulation	enhanced eIF4A activity and translation initiation
	p70-S6K1	increased phosphorylation	phosphorylation and inactivation of PDCD4 and eEF2K and enhanced translation initiation and elongation
Translation Elongation Factors	eEF2K	downregulation	enhanced activity of eEF2 and translation elongation
	eEF2	upregulation	enhanced translation elongation
Translation Initiation Factors	eIF4E	upregulation, increased phosphorylation at S209	enhanced translation initiation
	eIF4A1	upregulation	enhanced translation initiation
	eIF2α	upregulation, increased phosphorylation at S51	sequestration of eIF2B in an inactive complex, thereby limiting high translation rates
	eIF2B complex	upregulation	enhanced complex formation with p-eIF2α
Stress-related Kinase	GCN2	increased activity	increased phosphorylation of eIF2α

Summary of important ribosomal components, signaling proteins as well as translation factors, their deregulation and impact on protein synthesis in CRC.

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
