# Peer review of "Targeting Protein Synthesis in Colorectal Cancer"

_cancers, 2020, doi:10.3390/cancers12051298_

Round 1

Reviewer 1 Report

The review by Schmidt and co-workers addresses the interesting issue of targeting protein synthesis deregulation as a therapeutic strategy for CRC. Despite promising results in cell and animal models however, only a few developed strategies advanced into clinical trials, and most revealed only limited efficacy.

The authors provide a remarkable and comprehensive body of information on the main mechanisms leading to alterations of protein synthesis in CRC, presented in the format of an exhaustive list of reports for each paragraph. However, very few words have been spent in relating the information provided with steps of cellular transformation and tumor progression, and to discuss how changes in mRNA translation occurring in cancer affect cellular processes relevant to tumorigenesis.

Major issues

The introduction could be improved by providing a larger and less detailed view on the role of alterations of protein synthesis by briefly mentioning other possible mechanisms that are not specifically the focus of the review. It would also be interesting to better describe the potential role of alterations of protein synthesis in cancer susceptibility and whether they are observed in subjects at higher risk of developing CRC as inflammatory bowel disease patients or obese subjects prior to cancer development.

The very detailed paragraphs describing the process of protein synthesis could be simplified and more directly related to CRC.

Table 1 is confusing and the information shown does not always match with the text. As an example in the first line of the Table concerning ribosomal proteins, the deregulation in CRC is described as “upregulation, downregulation, mutation, deregulation of RNA pol II activity”. How this fits with the described overexpression of ribosomal protein genes, MYC-induced transcription of polymerases and hyperactivation of rRNA synthesis?

In the discussion, the authors should outline the questions that remain to be addressed to clinically advance the potential of therapeutic strategies targeting protein synthesis alterations. In light of the conclusions made by the authors on the limited success of clinical trials targeting protein synthesis alterations, this information should be anticipated in the abstract that highlights only the promising results in  cell lines, intestinal organoids, and mouse models.

Overall this review needs to be better structured to provide the readers with clear-cut take home messages. In particular, the review would benefit in clarity if a net distinction between alterations of each step of protein synthesis observed in CRC would be related to specific aspects of tumorigenesis, response to therapy or cancer susceptibility. On the same line, a separate description of the oncogenic pathways deregulated in CRC that affect specific components of the protein synthesis machinery would render easier text reading. In general, it should be avoided just to list the different reports quoting the main findings without amalgamating them in a thread.

Minor points

The legend to the figure has been included at the end of paragraph 4.3

Reviewer 2 Report

This is a clearly written review that covers thoughtfully the current knowledge on the role of deregulated protein synthesis in colorectal cancer and associated therapeutic strategies. While several in-depth reviews can be found in this field of translational control in cancer, this one has an advantage in that it focuses on colorectal cancer. No doubt it will be of interest to a wide audience.

I have one comment though. In addition to KRAS it might be worth mentioning BRAF mutation (BRAF V600E in particular) which occurs in about 10% of patients with metastatic CRC. Indeed, MAPK pathway upregulates mTORC1 activity: MEK1/2 promotes Raptor phosphorylation through ERK1/2 and p90 ribosomal S6 kinase.

Reviewer 3 Report

            The manuscript entitled “Targeting protein synthesis in colorectal cancer” by Scmidt et al. contains a review on an interesting subject within the field of basic oncological research. The authors focus on how translation occurs and what the peculiarities of this process in colorectal cancer (CRC) are. Most of the questions dealt with are only occasionally dealt with in other published reviews and this circumstance is of great value.

            The manuscript is well organised. After a brief introduction, the authors devote a brief section to reviewing the mechanisms of regulation of initiation, elongation and termination of the polypeptide chain. They afterwards review the deregulation of these processes in CRC and how they may offer potentially drugable targets. A final section deals with some clinical advances in this field.

            The manuscript is well written and authoritative. In fact, the authors have previously made some important contributions in the field. It meets the requirements to constitute a good review article, namely, the literature cited is comprehensive and up-to-date; the text is well organised; the authors do not limit themselves to collect a series of data, which are rather presented in a critical way. Future directions for research, although not given in a separate section, are distributed along the manuscript (e.g., lines 343 ff., 377 ff.).

            In some parts of the manuscript the large amount of data make it difficult to get a comprehensive view of the mechanisms described, especially when read by a non-specialist researcher. This occurs, for instance, in the section 2, “Mechanisms of regulation of protein synthesis”. The section is, of course, essential to centre the whole review and I think an early reference to Fig. 1 at this point may facilitate following the written description. In fact, all the mechanisms mentioned in section 2 are depicted in Fig. 1, and, in my opinion, there is no inconvenient in placing it within section 2. A caveat might be added in the figure caption mentioning that the targeting possibilities will be discussed later. In a similar way, I think it would be better to place table I at the end of section 3, as a summary of all the processes that are deregulated in CRC. In my opinion, an early inclusion to the table does not imply relevant advantages.

            A minor question concerns the sentence in line 237. If I am right, both eEFs and eIFs are deregulated in CRC, so the first words. “In contrast” may be misleading.

Round 2

Reviewer 1 Report

Upon revision the review has been substantially improved and all concerns have been properly addressed